# The Fecundity Characteristics and Spawning Strategy of *Uroteuthis edulis* in the East China Sea

**DOI:** 10.3390/ani13172786

**Published:** 2023-09-01

**Authors:** Rongpei Guo, Nan Li, Zhou Fang

**Affiliations:** 1College of Marine Sciences, Shanghai Ocean University, Shanghai 201306, China; 15038871631@163.com (R.G.); d210200060@st.shou.edu.cn (N.L.); 2National Engineering Research Center for Oceanic Fisheries, Shanghai Ocean University, Shanghai 201306, China; 3Key Laboratory of Sustainable Exploitation of Oceanic Fisheries Resources, Ministry of Education, Shanghai Ocean University, Shanghai 201306, China; 4Key Laboratory of Oceanic Fisheries Exploration, Ministry of Agriculture and Rural Affairs, Shanghai 201306, China; 5Scientific Observation and Experimental Station of Oceanic Fishery Resources, Ministry of Agriculture and Rural Affairs, Shanghai 201306, China

**Keywords:** *Uroteuthis edulis*, East China Sea, fecundity, dorsal mantle length, body weight

## Abstract

**Simple Summary:**

As one of the important economic fishery resources in the East China Sea, *Uroteuthis edulis* has attracted extensive attention from scholars around the world, but there is no relevant research on the fecundity characteristics and spawning strategies of *U. edulis* in the East China Sea. Therefore, this study takes the fecundity as the starting point to explore the fecundity characteristics and spawning strategies of *U. edulis*. Our study showed that the ovary oocytes of *U. edulis* mature in batches, after which the eggs are laid in batches. The dorsal mantle length (*ML*), water temperature at 25 m depth (T_25_) and sea surface height (SSH) are important indicators to measure the fecundity of *U. edulis*. The results of this study provide important fecundity information for understanding the population dynamics of *U. edulis* under the changing marine environment, and have important significance for the sustainable development and utilization of fishery resources.

**Abstract:**

The fecundity characteristics and spawning strategy of *Uroteuthis edulis* in the East China Sea were investigated by observing the potential fecundity (*PF*), relative fecundity by dorsal mantle length (*PF_ML_*) and relative fecundity by body weight (*PF_BW_*). The relationship between fecundity and a single biological indicator was measured, and generalized additive models (GAMs) were fit by adding environmental variables to help better understand this comprehensive relationship. The long diameter and short diameter of the ovarian oocytes ranged from 0.72 mm to 4.74 mm and from 0.46 mm to 3.67 mm, respectively. The long and short diameters of oviducal eggs ranged from 0.61 mm to 5.12 mm and from 0.39 mm to 3.81 mm, respectively. The egg diameter had a unimodal distribution. The *PF*, *PF_ML_* and *PF_BW_* ranged from 540 to 13,743 cells, 5 to 86 cells/mm and 6 to 53 cells/g, respectively. Three fecundity indicators were unimodally distributed, and the *PF_BW_* was more stable than the *PF_ML_* (*δ*^2^*PF_BW_* < *δ*^2^*PF_ML_*). The fecundity and single biological indicators were fitted, and it was found that the *PF* and *PF_ML_* were positively correlated with dorsal mantle length (*ML*) and body weight (*BW*). The generalized additive model (GAM) fitting showed that when considering the interaction between dorsal mantle length and sea surface height (M_13_), the deviation explanation rate of the *PF* and *PF_ML_* was the highest. Studies have shown that the ovary oocytes of *U. edulis* mature in batches, and then the eggs are laid in batches. The dorsal mantle length (*ML*), water temperature at 25 m depth (T_25_) and sea surface height (SSH) are important indicators to measure the fecundity of *U. edulis*. These findings allow for a deeper understanding of the *U. edulis* population dynamics for the future management of this economically and ecologically important species.

## 1. Introduction

Reproduction biology is an important part of life history research, and is of great importance to resource conservation, exploitation and utilization [1]. The long-term evolution of organismal reproductive strategies mainly refers to the ability to adapt in unique environments while improving individual reproductive performance such that the species’ survival and quantity of offspring are maximized [2]. The same is true of cephalopods (except for Nautilus), which have short lifespans and most only reproduce once. Meanwhile, their breeding strategy has high species specificity, including environmental adaptability [3]. The reproductive strategies of different cephalopod species have been formed through natural selection, which helps ensure the maximum adaptability of populations in their living environments. Research on cephalopod reproductive characteristics is necessary to understand their population variations, and to effectively manage and conserve cephalopod fishery resources.

At present, the East China Sea has become one of the main cephalopod fishing grounds globally. The estimated available number of cephalopods in the East China Sea is 21.2 × 10^4^ t, and the annual catch can reach 16 × 10^4^ t [4], which permits high exploitation and utilization. Since the collapse of the *Sepiella maindroni* resource in the 1990s, *Uroteuthis edulis* has gradually become an important economic species in the East China Sea [5]. With the increasing fishing intensity of *U. edulis*, the annual fishing effort could reach 1.5 × 10^4^ t, as its yield has increased year after year in the East China Sea [5].

The Kuroshio and Taiwan warm currents and the Yangtze River flushing fresh water and Yellow Sea cold water mass all affect the complex hydrological environment of the East China Sea. This environment has led to high primary productivity, making the East China Sea an important and popular natural fishing ground in China, and one of the world’s most highly productive waters [6]. However, the East China Sea’s changing marine environment (e.g., water temperature, salinity, chlorophyll) may affect the growth, reproduction and harvest of *U. edulis* [7]. For example, (i) the sea surface temperature affects population distributions, embryo development via incubation and reproductive time [7,8]; (ii) salinity extremes (high or low) can cause a significant decrease in embryo survival [9]; (iii) the mixed layer depth and chlorophyll-a concentrations can affect food distribution and abundance; and (iv) sea surface height can affect marine nutrient transport [10]. Additionally, the East China Sea is one of the fastest-warming large marine ecosystems in the world’s oceans, and this warming may make the sea temperature a crucial abiotic factor affecting the growth and reproduction of *U. edulis* [11].

*U. edulis* is a coastal warmwater cephalopod with fast generation renewal, rapid growth and a short life cycle [12,13]. This species is widely distributed in subtropical and tropical areas of the northwest Pacific Ocean, and in the coastal waters of eastern Africa and northern Australia [5]. The population structure of *U. edulis* is relatively complex, and it can be divided into spring, summer, autumn and winter-spawning groups, depending on their spawning season [12]. The growth and development of each group is affected by different environmental factors [14], making each group’s reproductive strategy and incubation temperature vary. In general, cold-water hatchlings tend to have larger body sizes and faster growth rates [15]. However, additional abiotic variables in hydrological environments can play key roles in hatchling success [7].

As one of the most economically important fishery resources in the East China Sea [16], *U. edulis* has attracted extensive attention from scholars globally. Studies have found that *U. edulis* in the southern East China Sea (subtropical waters) can lay eggs throughout the year, their life span is approximately 9 months, and their individual size and growth rates differ with season [17,18]. Li et al. [19,20] suggested that the East China Sea spawning peak of *U. edulis* occurs during spring and summer, and that these groups spawn in both southern and northern waters where incubation temperatures are 19–24 °C and 18–22 °C, respectively. Autumn spawning groups in the East China Sea have been found to migrate from the southern to the northeastern waters, and then migrate to the southern continental shelf to feed during spring and summer [14,17]. However, a relevant study on the fecundity characteristics of *U. edulis* in the East China Sea is absent from the peer-reviewed literature. Fecundity is directly related to recruitment and survival, which are crucial indicators to understand when determining the ability of a species to survive and maintain their populations under different management regimes [21]. Strengthening our knowledge of *U. edulis* fecundity is valuable as we attempt to correctly assess variations in the population biomass throughout the East China Sea, which would allow for rational and sustainable utilization and development of this fishery resource.

Therefore, to understand the reproductive biological characteristics of *U. edulis* in the East China Sea, squid samples were collected during the fishing season, and the hatching time was calculated according to the age of the statolith, allowing for the determination of the spawning group. This study focused on the fecundity characteristics and their comprehensive relationship with biological indicators and environmental factors, which provide important fecundity information for *U. edulis* to help better understand the population dynamics under the changing oceanographic environment.

## 2. Materials and Methods

### 2.1. Sampling U. edulis

Individual *U. edulis* samples were collected from a commercial trawler vessel (Zhelingyu 23860) in the East China Sea (27.5°–31° N, 123°–127° E) from November 2021 to March 2022 and September 2022 to March 2023. The samples were immediately frozen and transported to the laboratory for subsequent workup and analysis. After defrosting at room temperature in the laboratory, the 271 females sampled were used for basic biological determination, and then 102 females were selected for reproductive biology determination and calculation.

### 2.2. Biological Determination

Basic biological assays were performed on all defrosted samples in the laboratory, including gonadal maturity, dorsal mantle length (*ML*; nearest 1 mm), fin length (*FL*; nearest 1 mm), body weight (*BW*; nearest 1 g) and eviscerated weight (*EW*; nearest 1 g). The sex was determined by gonadal appearance, and gonadal maturity was divided into five stages (I, II, III, IV, V) according to gonadal development, where gonadal maturity in stages III and IV was the mature stage, and Ⅴ was the spawning stage [22,23]. Individuals with high gonadal maturity (III, IV) were selected for reproductive biology tests that measured their ovarian weight (*OW*; nearest 0.1 mg) and oviducal complex weight (*OCW*; nearest 0.1 mg).

### 2.3. Calculation of Hatching Dates and Determination of Spawning Groups

The statoliths were extracted from statocysts, and then rinsed with alcohol to remove organic matter and soft membranes from the statolith surface [24]. Statolith embedding, grinding and polishing were carried out according to methods from Liu et al. [25]. The prepared statoliths were photographed under an Olympus optical microscope, and Photoshop software was used to overlay the images. The age count was based on the rule of “one round per day” for cephalopods [26]. The age range for all of the samples was 170–246 d. According to the estimated hatching time based on age, the hatching of these individuals occurred from April to August, where the peak period was April, June and August. Therefore, the *U. edulis* samples in this study were mainly composed of spring and summer spawning groups.

### 2.4. Ovarian Oocyte Counts and Length Measurements

Subsamples of ovaries were selected from the anterior, middle and posterior areas in the carcass cavity. The ovarian subsample weights ranged from 0.15 to 0.30 g, and these weights were measured to the nearest 0.1 mg. The oocytes from the anterior, middle and posterior subsamples were dispersed in ultrapure water before counting in a 30 mm glass Petri dish. The oocytes were counted under a dissecting microscope (1×~4.5×), and the total count of the ovarian oocyte number (*OON*) was calculated by combining the three subsamples for each individual. After the oocyte count was completed, 15–50 oocytes were randomly selected depending on the OON, and photos were taken under a dissecting microscope equipped with a digital camera (1×~4.5×). ImageJ software was then used to measure the major axis of each oocyte long length (*OLL*) and the major axis of each oocyte short length (*OSL*) to 0.01 mm. The average of the OLL and OSL for each individual *U. edulis* was taken.

### 2.5. Oviducal Egg Counts and Length Measurements

According to the position of the oviduct in the abdomen of the carcass cavity, three subsamples were selected from the middle section of the cavity for each individual. The subsample weights ranged from 0.15 to 0.30 g, and were measured to the nearest 0.1 mg. The count of oviducal eggs was determined as explained above, and the oviducal egg number (*OEN*) was calculated by combining the three subsamples for each individual. After the egg counts were completed, 15–50 oocytes were randomly selected as described above. The major axis of the egg long length (*ELL*) and the major axis of the egg short length (*ESL*) were measured to the nearest 0.01 mm, and the measurements were averaged for each individual.

### 2.6. Data Analysis

The indicators related to the female reproductive characteristics of *U. edulis* were analyzed, including potential fecundity (*PF*; the sum of ovarian oocytes and oviducal eggs represented as cells) [27], relative fecundity by dorsal mantle length (*PF_ML_*; the ratio of potential fecundity to the dorsal mantle length [cells/mm] of the corresponding individual), and relative fecundity by body weight (*PF_BW_*; the ratio of potential fecundity to the body weight [cells/g] of the corresponding individual) [27]. The relevant calculation formulas are as follows:*PF* = *OON* + *OEN*(1)
(2)$${PF}_{ML}=\frac{PF}{ML}$$
(3)$${PF}_{BW}=\frac{PF}{BW}$$

In this study, four mathematical models of linear, exponential, logarithmic and power functions were used to fit the individual fecundity (*PF*, *PF_ML_*, *PF_BW_*) of *U. edulis* with a single biological index, and the minimum value of Akaike’s information criterion corrected (AICc) was used to determine optimal fitting. Then, the generalized additive model (GAM) was used to fit the relationships among individual fecundity, main biological indicators such as *ML* and *BW*, and environmental factors such as the water temperature at 25 m depth (T_25_) and the sea surface height (SSH). Before constructing the GAM model, the variance inflation factor (VIF) was used to test the collinearity of the explanatory variables, and when the VIF < 10, it was considered that there was no serious multicollinearity between the variables [28]. The GAM model is expressed as follows:(4)$$Y=\alpha +\sum_{i=1}^{n}f_{i}\left(x_{i}\right)+\varepsilon$$
where α is the intercept in the fitting model; *x_i_* is the explanatory variable; *f*(*x_i_*) is an arbitrary univariate smoothing function of each explanatory variable; and *ε* is the residual, and follows a normal distribution.

The nonlinear relationship between the response variables (fecundity indicator transformed by natural logarithms, such as ln (*PF*), ln (*PF_ML_*), ln (*PF_BW_*)) and the explanatory variables (*ML*, *BW*, T_25_, SSH) was constructed, and the GAM model followed a Gaussian distribution. The interaction between explanatory variables was considered when fitting the model, and different combinations of explanatory variables were added into the expression to obtain different GAM models. Akaike’s information criterion (AIC) [29] was used to test the fitting effects of different GAM models. The smaller the AIC value is, the better the model fitting effect [30]. The T_25_ and SSH data were downloaded from the Copernicus Marine Data Center website (https://data.marine.copernicus.eu/products), accessed on 20 April 2023. All statistical analyses were performed using Excel 2021 and R 4.2.2 [31].

## 3. Results

### 3.1. Ovarian Oocyte and Oviducal Egg Lengths

The long diameter and short diameter of the ovarian oocytes ranged from 0.72 mm to 4.74 mm (mean and standard error; 1.85 ± 0.80 mm) and from 0.46 mm to 3.67 mm (mean and standard error; 1.35 ± 0.60 mm), respectively. The long and short diameters of the oviducal eggs ranged from 0.61 mm to 5.12 mm (mean and standard error; 1.85 ± 0.78 mm) and from 0.39 mm to 3.81 mm (mean and standard error; 1.36 ± 0.58 mm), respectively. In addition, the size distribution of the ovarian oocytes and oviducal eggs showed a unimodal distribution (Figure 1a–d), in which the long diameter of the oocytes and eggs ranged from 1.2 mm to 1.8 mm (accounting for 32.69% and 36.93% of variation, respectively). All of the short diameters ranged from 0.8 mm to 1.2 mm (accounting for 38.39% and 37.98% of the variation, respectively).

### 3.2. Fecundity

The potential fecundity (*PF*), relative fecundity by dorsal mantle length (*PF_ML_*) and relative fecundity by body weight (*PF_BW_*) ranged from 540 to 13,743 cells (mean and standard error; 3474 ± 2623 cells), 5 to 86 cells/mm (mean and standard error; 23 ± 14 cells/mm) and 6 to 53 cells/g (mean and standard error; 25 ± 10 cells/g), respectively. The error comparison shows that the *PF_BW_* is more stable than the *PF_ML_* (i.e., *δ*^2^*PF_BW_* < *δ*^2^*PF_ML_*). The *U. edulis PF*, *PF_ML_* and *PF_BW_* were all distributed unimodally (Figure 2a–c). Among them, the peak ranges of the *PF*, *PF_ML_* and *PF_BW_* were 0–6000 cells (accounting for 87.25% of variation), 0–40 cells/mm (accounting for 89.22% of variation) and 10–40 cells/g (accounting for 89.22% of variation), respectively.

### 3.3. The Relationship between Fecundity and Biological Indicators

The relationship between the fecundity and a single biological index of *U. edulis* was fitted and analyzed using four mathematical models: linear, exponential, logarithmic and power functions. The model with the lowest AICc value was selected as the optimal model. The results showed that the *PF* of *U. edulis* had a power function relationship with the dorsal mantle length and a logarithmic function relationship with body weight. The *PF_ML_* had a linear relationship with the dorsal mantle length and a logarithmic relationship with body weight. Among them, the *PF* and *PF_ML_* were positively correlated with dorsal mantle length, fin length, body weight, net weight and ovarian weight. The *PF_BW_* showed a linear relationship with dorsal mantle length and body weight, while the *PF_BW_* was only positively correlated with fin length and ovarian weight (Table 1).

### 3.4. The Relationship between Fecundity and Biological Indicators and Environmental Factors

Regarding the *PF*, the simplest model, M_1_, has the lowest goodness of fit. Subsequently, considering four explanatory variables (*ML*, *BW*, T_25_, SSH) arranged in various combinations to fit the models (M_3_-M_15_), it was observed that the model with the interaction between *ML* and SSH (M_13_) has the highest explanatory power, followed by the model with the interaction between *ML* and T_25_ (M_12_). Regarding the *PF_ML_*, the model with the interaction between *ML* and SSH (M_13_) has the highest explanatory power, while the model with the interaction between *BW* and SSH (M_15_) has the second highest explanatory power. Regarding the *PF_BW_*, the model with the interaction between *BW* and T_25_ (M_14_) has the highest explanatory power, while the model with the interaction between *ML* and T_25_ (M_12_) has the second highest explanatory power (Table 2).

## 4. Discussion

### 4.1. Oocyte Length and Spawning Strategy

To some extent, the egg size reflects the reproductive and larval characteristics of marine organisms [32]. Large eggs may indicate that a population has good genetic characteristics, high survival rates, and high-quality reproductive offspring [33]. The study herein found that the long and short diameters of *U. edulis* oocytes were 0.72–4.74 mm and 0.46–3.67 mm, respectively, which were close to the egg sizes of other species of Loliginidae such as *Doryteuthis gahi* [34], *Loligo forbesi* [35], *Doryteuthis pealeii* [36], *Loligo vulgaris* [35], *Alloteuthis media* [37] and *Alloteuthis subulata* [38]. However, there are exceptions in the Loliginidae species, and previous studies have shown that mature eggs of species in the genus *Sepioteuthis* are relatively large, in the range of 5–10 mm egg diameters [39].

Cephalopods (except Nautilus) lay eggs once in a lifetime, with highly variable reproductive strategies. The spawning patterns of cephalopods show a single batch of simultaneous terminal spawning, multiple batches of intermittent spawning, or multiple spawning groups [40]. This flexibility in reproductive strategies could help a species cope with fluctuations in abiotic (e.g., temperature, fishing activity) and biotic (e.g., maternal ration, predator pressure) conditions [41]. The oviduct in the female reproductive system of *U. edulis* plays a role in temporarily storing mature eggs [42]. In this study, the number of mature eggs in the oviduct of *U. edulis* is far less than its potential fecundity, indicating that the ovarian oocytes cannot mature all at once, but mature in batches and then discharge in batches. The distribution pattern of ovarian oocytes and oviduct mature eggs of *U. edulis* showed a single peak interval, which also proves to an extent that the development pattern of ovarian oocytes is to mature in batches. Previous studies have suggested that there may be two spawning strategies in *U. edulis*: (i) Japanese spring spawning groups lay a small number of large eggs at a high spawning frequency, resulting in a relatively long spawning season, and (ii) summer spawning groups lay a large number of small eggs with a lower spawning frequency, and the spawning season ends quickly [43]. Small oocytes require more time to mature into large eggs, while larger oocytes take less time to grow into smaller eggs [44], further confirming the pattern of batch maturation of the ovarian oocytes of *U. edulis*.

### 4.2. Fecundity

Fecundity reflects an organism’s ability to reproduce offspring, which is a crucial indicator of population health and dynamics [45]. This study found that the *PF* of the *U. edulis* range was 540–13,743 cells, which was close to the *PF* of other Loliginidae species, such as *D. gahi* [34], *A. subulata* [38], *A. media* [37] and *L. forbesi* [35] (Table 3). However, exceptions include *L. vulgaris* [35] and *D. pealeii* [46], where the *PF* is relatively large, and the maximum *PF* can reach 110,000 cells. Additionally, the *PF* of *S. lessoniana* [47] was relatively small at only 180–1054 cells. This may be due to different environments, sampling areas and adaptations driving differences in the growth and development of individuals. The water temperature in the low-latitude sea area is high, food is abundant and the hydrologic environmental changes are continuously fluctuating and complex, which makes the *PF* of individuals vary greatly, whereas the opposite is true in high-latitude seas [48].

The *PF_ML_* and *PF_BW_* are important indexes to measure the fecundity of cephalopod individuals. The results of this study showed that the relative fecundity was better predicted by body weight than by dorsal mantle length, indicating that *U. edulis* has high reproductive investment in a larger single egg. In this study, the *PF_BW_* of *U. edulis* was 6–53 cells/g (mean 25 ± 10 cells/g), while other studies have shown that the *PF_BW_* was greater than that of *U. edulis* (e.g., *L. vulgaris* [35] and *A. media* [37]). These results indicate that *U. edulis* has fewer eggs but a greater volume of eggs, such that each egg has a higher probability of successfully developing into an adult. The difference in the *PF_BW_* of the above three species may be due to the complex ecological environments around them, such as the water temperature, food abundance, and habitat availability in different seas.

### 4.3. The Relationship between Fecundity and Biological Indicators

The fecundity of cephalopods usually correlates significantly with biological indicators such as dorsal mantle length and body weight, which can be expressed by several mathematical models. The most commonly fit models include linear, exponential, logarithmic and power functions. The relationship between fecundity and biological indicators of different cephalopods is also usually different. The *PF* of *Sepia officinalis* in the offshore area of Mauritania [27] experiences an exponential function of the dorsal mantle length and a power function of body weight, while the *PF* of *Sthenoteuthis oualaniensis* in the South China Sea [50] has a linear function relationship with the dorsal mantle length, which is different from the results presented here. Different cephalopod species have different fecundities and different correlations with biological indicators, which may be related to environmental factors, geographical location, nutritional status and individual reproductive characteristics.

The reproductive system of cephalopods is specialized, and is located at the posterior of the mantle cavity [51]. The size of the mantle determines the number of eggs carried by the ovaries and oviducts. Generally, the larger the dorsal mantle length, the higher the *PF*, and *U. edulis* showed a positive correlation between the two. Other studies have shown similar trends, such that the *PF* of Loliginidae species was positively correlated with carcass size (e.g., *L. vulgaris* [35], *L. forbesi* [35], *A. media* [37], *S. lessoniana* [47]) and body weight (e.g., *L. vulgaris* [35] and *S. lessoniana* [47]). These results showed that the larger the individual, the greater its fecundity. Another study showed that the reproductive system size is also one of the factors impacting fecundity [52]. The *PF* of female individuals of *L. vulgaris* [53] and *S. lessoniana* [47] were positively correlated with ovarian weight, which was consistent with the results of this study.

### 4.4. The Relationship between Fecundity and Environmental Factors

As a short-lived species, *U. edulis* is easily affected by its surrounding environment [54], and it lives in shallow coastal waters where its active water layer is mainly concentrated at a depth of 25 m [19]. Hence, in this study, the GAM model primarily selected the T_25_ and SSH at a depth of 25 m as key explanatory variables. A previous study showed that the autumn and winter fishing populations of *U. edulis* may be spawning populations in the spring and summer [23], which was consistent with the results of this study, and the distribution of spawning sites and migration routes of different spawning populations may be affected by the surrounding marine environment [14,18], resulting in differences in the reproductive characteristics of *U. edulis*. Among the GAM models, when the interactive effects of *ML* with the T_25_ and SSH were considered, the models (M_12_ and M_13_) showed relatively higher explanatory power, indicating that the growth and reproduction of individual *U. edulis* were more influenced by the environmental factors T_25_ and SSH.

## 5. Conclusions

In conclusion, the ovary oocytes of *U. edulis* mature in batches and the eggs are then laid in batches. The *ML*, T_25_ and SSH are important indicators to measure the fecundity of *U. edulis*. In addition, the fecundity of *U. edulis* was found to be more sensitive to changes in the marine environment. In this study, the T_25_ and SSH were selected as the environmental factors for analysis; however, other environmental factors (e.g., salinity and chlorophyll concentration) could be added to future research attempts to observe the effects of climate change on the fecundity and population dynamics of *U. edulis*.

## Figures and Tables

**Figure 1 animals-13-02786-f001:**
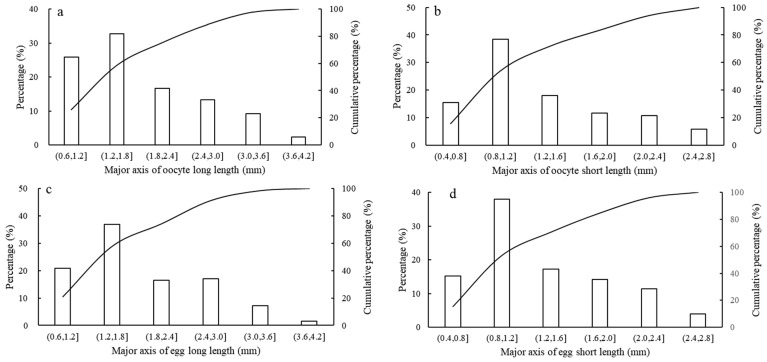
The ovarian oocyte size and oviducal egg size distributions of *U. edulis* ((**a**) represents major axis of oocyte long length, (**b**) represents major axis of oocyte short length, (**c**) represents major axis of egg long length, (**d**) represents major axis of egg short length).

**Figure 2 animals-13-02786-f002:**
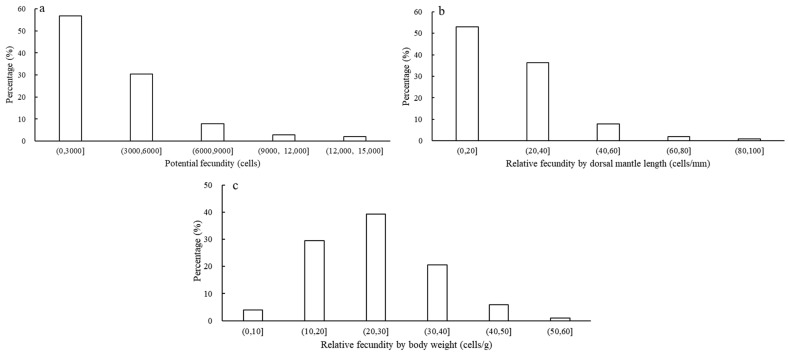
The frequency distributions of the potential and relative fecundities of *U. edulis* ((**a**) represents potential fecundity, (**b**) represents relative fecundity by dorsal mantle length, (**c**) represents relative fecundity by body weight).

**Table 1 animals-13-02786-t001:** Optimal equations between individual fecundity and single biological indicators of *U. edulis*.

Indicators	Biological Indicators	Individual Fecundity		
		Potential Fecundity, *PF* (Cells)	Relative Fecundity by Dorsal Mantle Length, *PF_ML_* (Cells/mm)	Relative Fecundity by Body Weight, *PF_BW_* (Cells/g)
Length indicators	Dorsal mantle length, *ML* (mm)	*PF* = 0.03*ML*^2.38^, *R*^2^ = 0.49	*PF_ML_* = 0.24*ML* − 10.76, *R*^2^ = 0.26	*PF_BW_* = 25.3 − 0.005*ML*, *R*^2^ = 0.0002
	Fin length, *FL* (mm)	*PF* = 84.03*FL* − 3565.33, *R*^2^ = 0.50	*PF_ML_* = 0.38*FL* − 8.20, *R*^2^ = 0.33	*PF_BW_* = 23.25 + 0.02*FL*, *R*^2^ = 0.001
Weight indicators	Body weight, *BW* (g)	*PF* = 3028ln (*BW*) − 11021, *R*^2^ = 0.56	*PF_ML_* = 14.1ln (*BW*) − 44.4, *R*^2^ = 0.40	*PF_BW_* = 27.15 − 0.02*BW*, *R*^2^ = 0.04
	Eviscerated weight, *EW* (g)	*PF* = 2614ln (*EW*) − 7007, *R*^2^ = 0.46	*PF_ML_* = 12ln(*EW*) − 24.9, *R*^2^ = 0.32	*PF_BW_* = 27.6 − 0.04*EW*, *R*^2^ = 0.06
	Ovarian weight, *OW* (g)	*PF* = 420*OW*^1.12^, *R*^2^ = 0.63	*PF_ML_* = 11.2exp^0.10^*^OW^*, *R*^2^ = 0.54	*PF_BW_* = 19.3exp^0.04^*^OW^*, *R*^2^ = 0.12

**Table 2 animals-13-02786-t002:** The generalized additive model of individual fecundity and biological indicators and environmental factors of *U. edulis*.

Model	Expression	Potential Fecundity, *PF* (Cells)			Relative Fecundity by Dorsal Mantle Length, *PF_ML_* (Cells/mm)			Relative Fecundity by Body Weight, *PF_BW_* (Cells/g)		
		R^2^	DE/%	AIC	R^2^	DE/%	AIC	R^2^	DE/%	AIC
M_1_	F~*ML*	0.61	61.2	126.65	0.35	36.2	126.34	0.04	6.98	116.21
M_2_	F~*BW*	0.72	73.5	98.13	0.51	52.2	100.30	0.18	20.3	100.44
M_3_	F~*ML* + *BW*	0.72	73.9	98.89	0.51	53.3	99.87	0.17	20.6	101.79
M_4_	F~*ML* + T_25_	0.62	62.7	124.52	0.38	38.9	124.03	0.06	10.2	114.09
M_5_	F~*ML* + SSH	0.65	67.3	118.80	0.39	40.5	121.94	0.06	9.04	114.82
M_6_	F~*ML* + BW + T_25_	0.72	74.9	97.15	0.52	54.6	99.03	0.19	23.1	100.00
M_7_	F~*ML* + BW + SSH	0.74	77.4	92.80	0.53	55.4	97.24	0.22	25.4	97.17
M_8_	F~*BW* + T_25_	0.72	74.4	96.88	0.52	53.7	98.89	0.20	22.9	99.05
M_9_	F~*BW* + SSH	0.74	76.7	93.16	0.53	54.9	96.95	0.22	25.2	96.12
M_10_	F~*BW* + T_25_ + SSH	0.74	77.8	94.16	0.53	55.1	98.68	0.22	26.1	97.07
M_11_	F~*ML* + *BW* + T_25_ + SSH	0.74	77.9	94.81	0.53	55.6	98.88	0.21	26.2	98.34
M_12_	F~*ML*: T_25_ + *BW* + SSH	0.75	78	90.73	0.55	58.3	94.54	0.26	30.8	93.69
M_13_	F~*ML*: SSH + *BW* + T_25_	0.75	78.7	92.33	0.56	60.9	95.69	0.23	28.2	97.19
M_14_	F~*BW*: T_25_ + *ML* + SSH	0.73	74.7	94.72	0.55	58.3	95.60	0.26	31.6	94.31
M_15_	F~*BW*: SSH + *ML* + T_25_	0.72	75.5	100.75	0.55	59.9	100.51	0.23	29.3	99.07

**Table 3 animals-13-02786-t003:** Egg sizes and fecundities for some Loliginidae species.

Species	Egg Size (mm)	Potential Fecundity, *PF* (Cells)	Relative Fecundity by Body Weight, *PF_BW_* (Cells/g)	References
*Doryteuthis gahi*	1.66–2.39	1100–21,000	-	[34]
*Loligo forbesi*	0.1–4.3	1000–16,000	-	[35]
*Doryteuthis pealeii*	2.2–2.6	5700–117,000	-	[36,46]
*Loligo vulgaris*	0.1–4.1	28,500–74,200	114–251	[35]
*Alloteuthis media*	1.5–2.3	900–4500	56–524	[37]
*Alloteuthis subulata*	1.7–1.9	2100–13,500	-	[38]
*Sepioteuthis lessoniana*	5.5–5.9 × 3.6–4.4	180–1054	-	[47,49]

## Data Availability

The data that support the findings of this study are available from the corresponding author upon reasonable request.

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
