# Peer review of "The Fecundity Characteristics and Spawning Strategy of Uroteuthis edulis in the East China Sea"

_animals, 2023, doi:10.3390/ani13172786_

Round 1
Reviewer 1 Report
This work is well-written and with important information on a species of high economic importance. All this information is relevant considering the exploitation levels this species is subject to.
However, I have detected some flaws that preclude me from recommending this work for publication in its present form.
For starters, no clear working hypothesis or research question guides this research. Therefore the scope of this work is unclear. What central question are the authors trying to answer in this research? This should be very clear.
Through this research, the authors took many samples and generate a high quantity of information, however, in the end, it was not used. For example, they estimated the gonadal maturity stages and condition factor, but all this information was not used except for the purpose of fecundity, and then the discussion focus almost exclusively on fecundity, using three different approaches, which seems to be the main focus of this research.
But even this is confusing. Are the authors trying to show the best way to analyze the fecundity of this species? Why were they trying to answer when analyzing the relationship between fecundity and biological indicators?
Table 1 is also very confusing. The text refers to the correlation of the models used, but this does not appear in the table.
Same as Table 2, which is also very confusing. Where do these equations come from?
The discussion and the conclusion are also confusing and difficult to understand. The reason t for this is the lack of a working hypothesis and clear research questions. Probably a more adequate title for this work should be the spawning strategy of U. edulis in the East China Sea, as the initial line of the conclusion states, as this work deals with the spawning strategy and not only the fecundity characteristics.
Reviewer 2 Report
In the manuscript “The fecundity characteristics of Uroteuthis eduli in the East China Sea”, the authors studied the fecundity characteristics of U. eduli and established several models to calculate the relationship between fecundity and biological indicators or environmental factors. The results of the present study provided meaningful information for understanding the population dynamics and the fishery management. However, several issues should be addressed before accepting of this manuscript.
1. The paragraph (Lines 62-76) is introducing some environmental characteristic of East China Sea, and it is better to switch its position with the above paragraph (Lines 50-61).
2. Line 118, what is “high gonadal maturity”. Detailed developmental stages should be provided.
3. Line 149, the abbreviation of oviducal egg number (OON) is the same to the ovarian oocyte number (OON)? Maybe OEN?
4. The full name of AICc should be given when it first appeared in line 168, and so were T25, SHH.
5. What is Figure 1, which is not introduced in the main text. And all the figure numbers cited in the main text are not correct.
6. The detailed information of models between fecundity and biological indicators and environmental factors are missing? I cannot find either the mathematical models established (M1, M2…), or the criterions to judge each model (AICc, explanatory power…). In addition, Table1 and Table 2 are not correctly cited in the main text, and what are Table 3 and Table 4?
7. I cannot obtain the conclusion that “U. edulis exhibited intermittent, multibatch spawning strategies” in section 4.1. More accurate discussion is needed.
8. Line 309, it is hard to understand the sentence “The relative fecundity of individual fish can be compared with the fecundity of different species of fish…”, which needs rephrasing.
The manuscript is well written. Only some small mistakes existed.
Round 2
Reviewer 1 Report
The authors have addressed all the concerns raised in the first review adequately.
The manuscript would benefit from a revision of a colleague whose first language is English
Author Response
Thanks for affirming this article.
Reviewer 2 Report
The author gave a detailed response to my query and made corrections for issues and deficiencies in the manuscript. Thus, the manuscript has reached the standard for publication.
Author Response
Thank you for affirming this article.